# Solar-driven selective conversion of millimolar dissolved carbon to fuels with molecular flux generation

Bin Liu[1,2], Zheng Qian[1,2], Xiang Shi[1,2], Haoqing Su[1,2], Wentao Zhang [1,2], Atsu Kludze [1,2], Yuze Zheng[1,2,3], Chengxing He [1,2], Rito Yanagi[1,2] & Shu Hu [1,2] ✉

The direct utilization of dissolved inorganic carbon in seawater for $CO_2$ conversion promises chemical production on-demand and with zero carbon footprint. Photoelectrochemical (PEC) $CO_2$ reduction ($CO_2R$) devices promise the sustainable conversion of dissolved carbon in seawater to carbon products using sunlight as the only energy input. However, the diffusion-dominant transport mechanism and the near-zero concentration of $CO_2(aq)$ ($CO_2$ dissolved in aqueous solution) in static seawater has made it extremely challenging to achieve high solar-to-fuel (STF) efficiency and high carbon-product selectivity. Here, where $CO_2(aq)$ as a reactant generated in situ by acidification of $HCO_3^-$ flows continuously from $BiVO_4$ photoanodes to Si photocathodes, enabling a single-step conversion of dissolved carbon into products. Our PEC device significantly increases the CO selectivity from 3% to 21%, which approaches the 30% theoretical limit according to multi-physics modeling. Meanwhile, the $Si/BiVO_4$ PEC $CO_2R$ device achieved a STF efficiency of 0.71%. Such flow engineering achieves flow-dependent selectivity, rate, and stability in simulated seawater, thus promising practical solar fuel production at scale.

Developing carbon capture, utilization, and storage technologies is crucial for managing anthropogenic carbon dioxide ($CO_2$) emissions while providing sources of sustainable chemicals and fuels. However, direct air capture is energy-intensive and costly[1,2]. In comparison, dissolved inorganic carbon in seawater has a bicarbonate molarity that is ~140 times higher than the 420 ppm of atmospheric $CO_2$. Furthermore, seawater is a natural carbon sink of net ~0.4 giga-ton $CO_2$ per year via the flux exchange between the seawater and atmosphere, potentially supporting trillion-ton-scale $CO_2$ capture, utilization, and sequestration via engineering solutions[3,4]. In particular, the conversion of dissolved carbon in seawater into carbon-based fuels or chemical products can be entirely powered by sunlight using photoelectrochemical (PEC) devices. Such solar-powered chemical devices, floating on the ocean, could utilize ocean current, tidal energy, and

sunlight to generate dissolved $CO_2$ on demand via bicarbonate acidification, i.e., $HCO_3^- + H^+ \rightarrow CO_2(aq) + H_2O$. Then, the $CO_2$ reduction ($CO_2R$) reaction converts the dissolved $CO_2$, i.e., $CO_2(aq)$, to gaseous carbon products, such as CO or $C_2H_4$, for subsequent synthesis of liquid fuels or chemicals. These PEC devices employ pairs of photoanodes and photocathodes designed to drive water oxidation and fuel-forming $CO_2R$ reactions, respectively, with sunlight as the only energy inputs[5–7]. In a stepwise fashion, $CO_2$ can be captured from dissolved carbon in seawater as the carbon source for the subsequent PEC $CO_2R$ conversion[8,9]. Despite established technology, stepwise capture and conversion have rarely been demonstrated due to the challenges mentioned below. Alternatively, an integrated PEC process for the direct utilization of dissolved carbon for solar fuel production is expected to consume much less energy and be more economically

[1]Department of Chemical and Environmental Engineering, School of Engineering and Applied Sciences, Yale University, New Haven, CT, USA. [2]Energy Sciences Institute, Yale West Campus, West Haven, CT, USA. [3]College of Chemistry and Molecular Engineering, Peking University, Beijing, China. ✉e-mail: shu.hu@yale.edu

viable than the stepwise approach such as PV-driven $CO_2$ capture and $CO_2$ electrolysis.

Figure 1a illustrates a conventional PEC device configuration for unbiased $CO_2$ reduction and water oxidation, which operates in static electrolytes. In this design, the anodes and cathodes are arranged back-to-back in a tandem configuration[10,11], which maximizes light absorption. Fuel-forming reactions, including $CO_2$ reduction and $H_2$ evolution, proceed at the cathode, and water oxidation reactions proceed at the anode. Because the electrolyte of the PEC device is static, diffusion is the dominant transport mechanism, subsequently generating an ionic current that is carried with $H^+$, $HCO_3^-$, and other major ionic species. As shown in Fig. 1c, acid-base reactions occur during the diffusional transport of $CO_2(aq)$, $H^+$, and $HCO_3^-$. The pH gradient, ion transport, and acid-base buffer reactions between the anodes and cathodes are the major transport losses. Seawater, with its ionic strength of ~ 0.7 M, is a good ionic conductor[12,13], which means the electrolyte resistance has minimal contribution to the transport losses under solar illumination. $CO_2$ reduction or $H_2$ evolution consumes $H^+$, and net produces $OH^-$. The pH gradient typically contributes to notable transport loss in the PEC device. Diffusion-dominant transport creates a pH gradient in the buffered electrolyte (Fig. 1c), e.g., from 10 to 4, for a current density of $0.5 \, mA/cm^2$ in a static 2 mM $HCO_3^-$ buffer electrolyte, which leads to a higher pH at the cathode and reducing local $CO_2(aq)$. The pH difference between the cathodes and anodes adds to the device's potential loss, which can be as high as 500 mV, due to pH-dependent thermodynamic potentials[14,15]. Because of this, the solar-to-fuel (STF) conversion efficiency for $CO_2$ reduction, or water splitting broadly, is low: e.g., photoabsorbers of $SrTiO_3/BiVO_4$, amorphous-$Si/BiVO_4$, and perovskite/$BiVO_4$ (with $BiVO_4$ the photocurrent-limiting absorber) have achieved STF efficiency of 0.08%[16], 0.43%[17], and 0.63%[18] respectively.

The conventional diffusion transport of $H^+$ and $HCO_3^-$ in a pH-neutral and buffered electrolyte leads to a high pH near the $CO_2R$ cathodes. As shown in Fig. 1c, $CO_2(aq)$ is constantly consumed by $OH^-$ near the cathode. As a result, there is insufficient reactant for PEC $CO_2R$. This continuous $CO_2$-$OH^-$ reaction (Fig. 1c) depletes the $CO_2(aq)$ concentration to near zero, eliminating selectivity for carbon products such as CO and $C_2H_4$. Thus, another challenge of utilizing seawater arises from the low concentration of dissolved carbon species including $HCO_3^-$ and $CO_2(aq)$. Among these species, $HCO_3^-$ is not considered an active species during (photo-)electrochemical $CO_2$ reduction. However, seawater contains only 2.3 mM bicarbonate, and the concentration of electrochemically active $CO_2(aq)$ species is near zero. To utilize seawater for liquid solar fuel production, one typically takes a stepwise approach of first acidifying seawater to convert the $HCO_3^-$ to pure $CO_2$ and then bubbling the pure $CO_2$ continuously into a conventional PEC device to increase the reactant concentrations. To improve the selectivity for carbon products, it is necessary to ensure sufficient $CO_2(aq)$ reactants by constantly bubbling $CO_2$ gas. However, this approach is costly and is limited by low energy and carbon utilization efficiency (continuous $CO_2$ bubbling and $CO_2$-$OH^-$ reaction), and thus not economically viable. In addition, it is impractical to scale such $CO_2$ bubbling schemes for large-area PEC reactors situated on the ocean.

Overall, the combined challenge in reaction engineering and $CO_2R$ catalysis motivates new ways of effectively managing diffusion-based transport losses and performance degradation due to the depletion of $CO_2(aq)$ local to photocathodes. Thus far, reported PEC devices that leverage flowing seawater with 2.3 mM bicarbonate and without $CO_2$ bubbling remain largely unexplored. In this case, flow replaces diffusion to dominate the transport. Besides, the complex multi-species composition and chemical equilibrium of seawater is a notable challenge for direct ocean $CO_2$ capture and conversion, including major mineral species, i.e., $Na^+$, $Cl^-$, $Mg^{2+}$, and $Ca^{2+}$ [8,19], and organic carbon and microbial species. Chemical selective coatings are

needed to permeate active reactants and prevent catalyst poisoning by minerals or other impurities in seawater[20]. For proof-of-concept, simulated seawater was used.

We demonstrate an understudied regime of molecular flux catalysis in which $CO_2(aq)$ reactants continuously flow to the catalysts within a boundary layer flow (Fig. 1b and d). This boundary layer flow is generated by a custom-designed and 3D-printed PEC device, which has tandem pairs of $BiVO_4$ photoanodes to Si photocathodes that are arranged in parallel (Fig. 1b). PEC $CO_2R$ catalysis coupled with the boundary flow transport enables the conversion of the naturally low concentration of dissolved inorganic carbon found in seawater (2.3 mM $HCO_3^-$) to a flux of $CO_2(aq)$ (Fig. 1d). In this work, we make four parallel configurations for the Fig. 1b design, which in practice can be repeated periodically to achieve scaled-up, solar-driven, direct ocean $CO_2$ conversion. The boundary-layer shear flow confines a continuous flux of $CO_2(aq)$ molecules near the surface from the photoanode to the photocathode. The $BiVO_4$ photoanode generates $H^+$ thus acidifying $HCO_3^-$ in flowing seawater under sunlight. The flow also prevents the out-diffusion of the $CO_2(aq)$ to the bulk seawater, increasing the transport flux of $CO_2(aq)$ and $H^+$ and simultaneously reducing the $CO_2(aq)$'s residence time to achieve its efficient utilization at each photocathode downstream. Notably, the flow energy can be readily harvested from tidal waves and ocean currents, making such a boundary layer flow scheme possible.

The $CO_2(aq)$ flux, in situ, generated by the $BiVO_4$ photoanode, arrives at the catalyst of the Si photocathode and then gets converted to gaseous CO products which accumulate at the device headspace. Besides the $CO_2(aq)$ flux, the $H^+$ flux is enhanced by the flow. This boundary layer flow reduces the pH gradient (supported by Fig. 5 data) and other concentration-limited transport losses (supported by Fig. 6 simulation) to improve PEC device efficiency of $CO_2$ reduction, water splitting, and other fuel-forming photocatalysis broadly. Compared to conventional static PEC devices, the STF efficiency of our flow device increased from 0.4% to 0.71%, which is the highest efficiency among reported $BiVO_4$-based $CO_2$-reduction devices. This molecular flux approach tunes the $CO_2R$ catalytic selectivity via flow velocities. We also conducted modeling and numerical simulations to quantify the effect of flow vectors on the selectivity and rates of the PEC device. According to the simulation and experimental outcome, this molecular flux catalysis approach through boundary layer flow drastically increased the selectivity of carbon-based products such as CO from 3% to 21%—approaching the theoretical limit of utilizing seawater under the solar flux, whereas conventional static device (Fig. 1c) requires constant bubbling of $CO_2$ gas to achieve a comparable selectivity level. Resources such as seawater, air, and waste streams are typically low in the active species of $CO_2(aq)$ but high in bicarbonates and carbonates due to their much higher solubility. This molecular flux catalysis concept has the potential to enable the solar-driven direct conversion of low-concentration reactants or improve the utilization efficiency of dilute reactants.

## Results and discussion

### Design and construction of seawater flow reactor for $CO_2$ capture and in situ conversion

Simulated seawater was prepared by dissolving 35.5 g of simulated seawater salt (Instant Ocean®, see Table for detailed chemical compositions) in 1 L 18 mega ohm deionized water without $CO_2$ bubbling. The chemical composition of this simulated seawater solution was designed to simulate the chemical, electronic, and ionic properties of natural seawater (Supplementary Table S1). Although $Na^+$ and $Cl^-$ ions are the primary carriers of the ionic current in this solution, it is essential for protons generated at the upstream photoanodes to be transported to the downstream photocathodes for consumption. To address these issues, we developed an original vortex reactor design and 3D-printed the flow reactor (Fig. 2a, photographs in

**(a)  Catalysis in static solution with CO₂ purge**

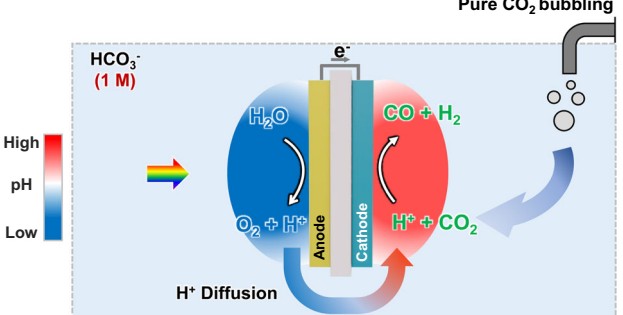

**(b)  Catalysis with CO₂ molecular flux**

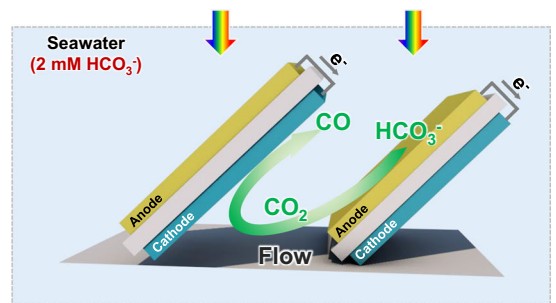

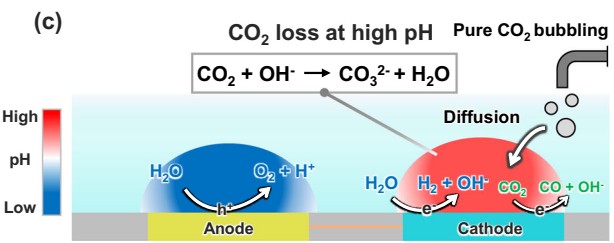

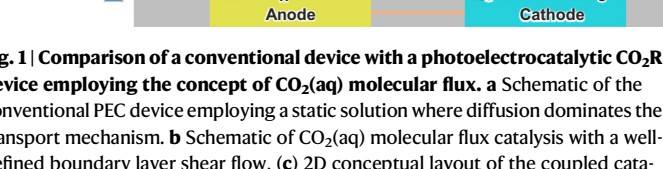

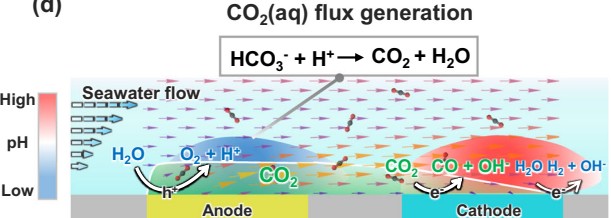

**Fig. 1 | Comparison of a conventional device with a photoelectrocatalytic CO₂R device employing the concept of CO₂(aq) molecular flux. a** Schematic of the conventional PEC device employing a static solution where diffusion dominates the transport mechanism. **b** Schematic of CO₂(aq) molecular flux catalysis with a well-defined boundary layer shear flow, (**c**) 2D conceptual layout of the coupled catalysis, acid-base buffer reaction, and diffusion processes between anodes and cathodes, (**d**) 2D conceptual layout of the molecular flux catalysis, acid-base buffer reaction, and convective flow processes from the anode to the cathode along the flow pathway. The arrows in Fig. 1d represent $CO_2$ flux, and the light green area represents the spatial distribution of $CO_2$ produced at the anode underflow. The pH ranges in (**c** and **d**) are derived from the pH map and modeling in Fig. 5.

Supplementary Fig. S1a and S1b). This 3D-printed reactor produces a boundary layer flow around tandem pairs of $BiVO_4$ photoanodes to Si photocathodes (Fig. 1b). This shear flow is distinctive to the vertical convective flow generated by, e.g., rotating disk electrodes. It enables the efficient conversion of the naturally low concentration of dissolved inorganic carbon (2.3 mM $HCO_3^-$) found in seawater to a flux of $CO_2(aq)$ molecules (Fig. 1d).

The 3D-printed vortex reactor features a periodic array of photoanode-photocathode pairs, each configured back-to-back and aligned in parallel. In total, four pairs of Si/$BiVO_4$ tandem photoelectrodes were mounted in our flow PEC reactor (Supplementary Fig. S1d). In each pair, a Si photocathode and a $BiVO_4$ photoanode are positioned back-to-back, maintaining a distance of 2 cm apart. The $BiVO_4$ photoanode absorbs the short-wavelength portion of the solar spectrum (300–520 nm) and the Si photocathode absorbs the long-wavelength light (520–1100 nm, Supplementary Fig. S2). The fraction of the incident radiation absorbed by the $BiVO_4$ and the Si is approximately 60% and 90%, respectively. The substrate where $BiVO_4$ deposits need to be transparent to allow light to pass through. Each $BiVO_4$-Si pair is oriented at an angle of 45° toward the sun and arranged horizontally in parallel to optimize light capture (Supplementary Fig. S1b). When the tilt angles are 30° (Supplementary Fig. S3a) and 60° (Supplementary Fig. S3b), respectively, their flow fields and velocity are similar to those observed at 45° (Fig. 3), indicating that the well-defined flow can bring in situ generated $CO_2(aq)$ from photoanode to photocathode. In principle, these angles can be adjusted to the latitude of reactor deployment by design, to maximize light capture. Ag-Au/$CrO_x$ co-catalysts are deposited on the Si photocathode for $CO_2R$ (fabrication for metallic catalysts and $CrO_x$ coatings in the "Methods" section). This design, where parallel pairs of photoelectrodes are utilized, can be indefinitely repeated for manufacturing scale-up. The holes are transported to the $BiVO_4$ surface to participate in the water oxidation reaction, and the electrons are transported to Si through the

back contact to conduct $CO_2$ reduction (Supplementary Fig. S1c). Electrical contacts are made between the back-to-back photoanode and photocathode pairs, while the ionic currents are between the upstream photoanode and the downstream photocathode. In this case, the parallel arrangement and relative distance between upstream $BiVO_4$ photoanodes and downstream Si photocathodes do not change, thus not affecting the flow field. The seawater flow over each photoelectrode pair is identical.

Seawater flow was designed to sweep across the surface of $BiVO_4$ photoanode, carrying $H^+$, $HCO_3^-$, and $CO_2(aq)$ from the photoanode surface to downstream photocathodes that facilitate $CO_2(aq)$ adsorption at the Ag-Au/$CrO_x$ catalysts and the subsequent in situ $CO_2$ reduction (Fig. 2b). The seawater flow field is characterized by laminar seawater flow across the $BiVO_4$ photoanode surface, confirmed through the calculations of Reynolds numbers under seawater flow velocities of 0.16–0.77 m/s at a representative 0.3 cm distance above photoanodes (simulated and listed in supporting information). Consideration of operational flow velocities was also given to the time needed to generate $CO_2(aq)$ and its finite lifetime of [$CO_2(aq)$] decay back to the equilibrium concentration when its concentration exceeded its equilibrium concentration in the bulk solution. To solve this concern, a vortex flow is created for $CO_2(aq)$ to pass over Si photocathode surfaces (Fig. 2c) mounted and electrically wired to the back of a $BiVO_4$ photoanode (shown in supporting Supplementary Movie S1). This design enables the direct transport of the generated $CO_2(aq)$ to the Si photocathode without hindering light absorption. Gas bubbles are continuously emitted from photoelectrode surfaces (Fig. 2d, see supporting Supplementary Movie S2), underscoring the importance of well-defined convective flow for solar-driven ocean-based $CO_2$ capture and conversion. Moreover, this flow reduces the effective residence time of $CO_2(aq)$ generated in situ at the $BiVO_4$ photoanode to the Au-Ag cocatalysts supported on the Si photocathode: the excess of $CO_2(aq)$ can live long enough to arrive at the Si

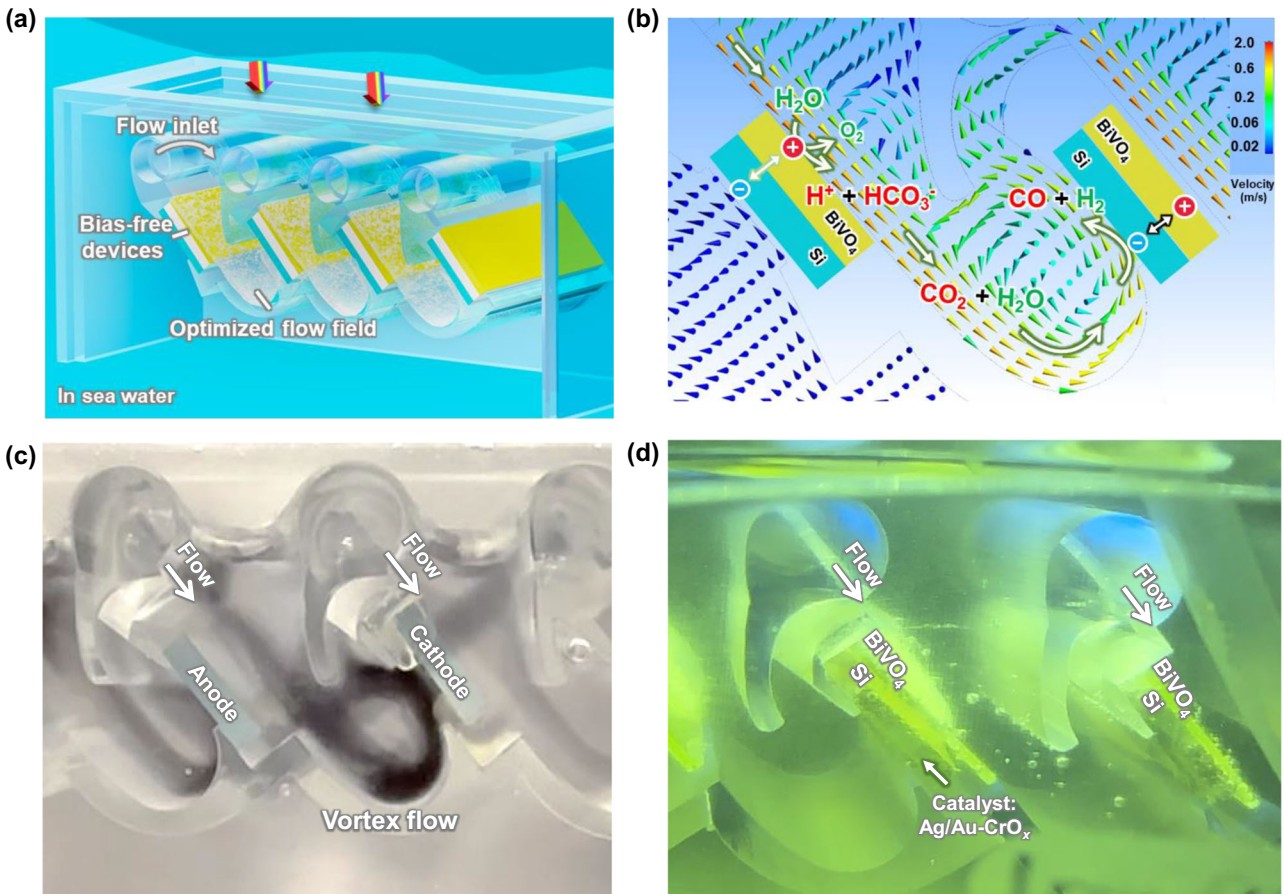

**Fig. 2 | Ocean reactor with enhanced flow field for $CO_2$ extraction and in situ utilization in seawater without bubbling $CO_2$. a** The schematic illustration of in situ generated $CO_2$ in a floating reactor in seawater. **b** The flow field simulation. **c** The vortex flow pattern is highlighted by black ink. **d** Photograph of $CO_2$ capture and in situ conversion in seawater under 1 sun AM 1.5 G. The x-axis in modeling aligns with the flow field direction. Si block indicates two side-by-side Si photo-absorbers connected in series to form a Si photocathode.

photocathode as active reactants and undergo subsequent $CO_2R$ conversion to valuable gaseous products before it reacts with alkaline species ($OH^-$, $CO_3^{2-}$) near the Si photocathode, or out-diffuses to the bulk seawater.

To effectively convert $CO_2$ to CO and suppress $H_2$ production, a 20 μm thick layer of Ag paste was uniformly scribbled to the surface of the Si substrate in direct contact with the indium tin oxide (ITO) layer (Supplementary Fig. S4), which serves as the catalytic and protective layer for Si photocathodes. To further improve the kinetics of $CO_2R$, a 5 nm Au layer was deposited onto the Ag layer surface, which is confirmed by XPS (Supplementary Fig. S5) and XRD (Supplementary Fig. S6). The $CrO_x$ layer was then deposited on the Au layer to prevent the oxygen reduction reaction and other side reactions in seawater. Therefore, light-driven $CO_2$ reduction on the photocathode was performed under AM 1.5 G illumination (100 mW/cm²) with $CO_2$ purge (pH 7) under ambient conditions. The Si photocathode with a single junction exhibits an onset potential of 0.3 V vs RHE (defined as the potential required to achieve a photocurrent of 0.1 mA/cm²) (Supplementary Fig. S4d). To further enhance the driving force for $CO_2$ reduction, double-junction Si photocathodes, which are connected with a Si solar cell in series, were fabricated (schematic in Supplementary Fig. S7, photograph in Supplementary Fig. S8). The onset potential was positively shifted to 0.75 V vs RHE, and the light-limited photocurrent is − 7.1 mA cm⁻². The performance showed a maximum applied bias photon-to-current efficiencies (ABPE) of 0.52% at 0.4 V vs RHE (Supplementary Fig. S9), which is among the best efficiency values for monocrystalline Si-based photocathodes (Supplementary

Table S5). The potential-dependent Faradaic efficiencies (FEs) toward CO and $H_2$ were evaluated, respectively, at various potentials. A considerable amount of CO was produced at 0.7 V vs RHE, at a FE of 40% for CO. The production of CO was achieved in a wide potential range from 0.7 to 0.1 V vs RHE (Supplementary Fig. S4e), demonstrating the great selectivity of Ag-Au catalyst layers and $CrO_x$ to eliminate the $O_2$-reduction side reactions. Therefore, the Si photocathodes show decent $CO_2R$ activity and stability in seawater, which is crucial for practical $CO_2$ conversion.

To reveal the PEC $CO_2R$ performance differences between the conventional static-solution configuration and our molecular flux catalysis, the overall solar-to-CO conversion efficiency was investigated by systematically varying the flow velocities during the operation of seawater-flow PEC devices under AM 1.5 G illumination (100 mW/cm²). In this study, flow velocity was defined using the average volumetric rate at which seawater solution left the inlet nozzle, which was 0.3 cm away from the closest photoelectrode pair (Supplementary Fig. S1b). To quantify the flow velocity, the volumetric flow velocity was divided by the cross-sectional area of the tubing. The fuels produced in this system include CO and $H_2$ gases, with liquid products falling below the detection limit, as confirmed by nuclear magnetic resonance (NMR) spectroscopy (Supplementary Fig. S10). Analysis of the gas product composition at a flow velocity of 0 m/s, revealed that the Faradaic efficiency for CO and $H_2$ are 3% and 97%, respectively. No other carbon-based products are detectable. The low CO selectivity suggests that there is a preference for electrons to reduce $H^+$ to form $H_2$ under static flow conditions (Fig. 3a), where $[CO_2(aq)]$ is as low as

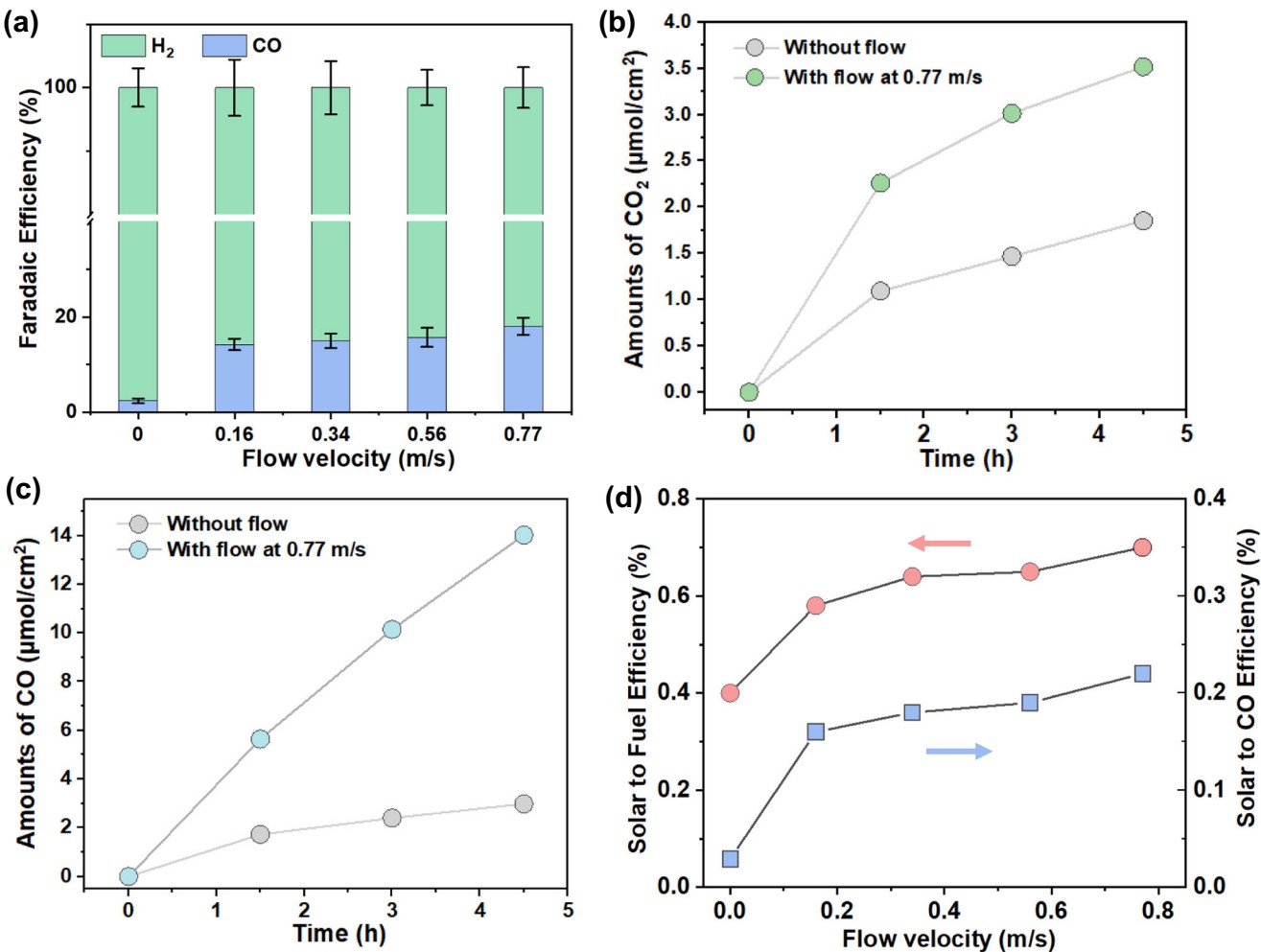

**Fig. 3 | Comparison of performance of extraction and in situ utilization of $CO_2$ from seawater at static and different flow velocities. a** The Faradaic efficiencies of CO and $H_2$ at static and various flow velocities. The error bars represent the standard deviations of three independent measurements of the same samples. **b** Time course of $CO_2$ gas production. **c** Time course of CO gas production. **d** Solar-to-Fuel and solar-to-CO efficiencies at different seawater flow velocities measured at 0.3 cm above the surface of the $BiVO_4$ photoanode near the inlet.

2 µmol (see modeling section). Increasing the flow velocity to 0.77 m/s, resulted in a CO Faradaic efficiency increase from 3% to 19%. This significant increase in CO production underscores the crucial role that flow plays in both the in situ generation and mass transfer of $CO_2$(aq) from the upstream photoanode to the downstream Si photocathode (Fig. 3a). Under conventional PEC configuration (the static solution), CO generation over a period of 4.5 h was 2.97 µmol per $cm^2$ illumination area. In contrast, CO generation over the same period significantly increased to 14 µmol/$cm^2$ when the flow velocity was increased to 0.77 m/s (Fig. 3c). The CO production rates at flow velocities of 0–0.56 m/s are shown in Supplementary Fig. S11a. The increase in CO production can be attributed to the well-fined flow field, which increases $CO_2$ concentration at the cathode.

A similar trend was observed when measuring gaseous $CO_2$ generation (Fig. 3b). Under static flow conditions, $CO_2$ generation was 1.85 µmol per $cm^2$ illumination area over a period of 4.5 h. When the flow velocity was increased to 0.77 m/s, the amount of extracted $CO_2$ over the same timeframe increased to 3.5 µmol/$cm^2$. Excess $CO_2$(aq) at the headspace-seawater interface may eventually be released as $CO_2$ gas. The rate for excess $CO_2$(g) accumulation at flow velocities of 0–0.56 m/s are shown in Supplementary Fig. S11b. Correspondingly, the STF efficiencies increased from 0.4% to 0.71% with the increase in flow velocity (Fig. 3d). This level of performance serves as our benchmark STF for $BiVO_4$-based devices for unbiased $CO_2$R

(Supplementary Table S6). The observed increase in $CO_2$ generation supports our hypothesis that in situ generated $CO_2$(aq) has a sufficient lifetime to adsorb at the Ag-Au/$CrO_x$ interface and participate in $CO_2$R downstream at the Si photocathode.

## Construction of $BiVO_4$ photoanode for $CO_2$ extraction and prevention of side reactions in seawater

The $BiVO_4$ photoanode was chosen because it can provide the high oxidation potential per hole charge transfer required for seawater oxidation. Thin-film $BiVO_4$ photoanodes (Fig. 4a) were fabricated on fluorine-doped tin oxide (FTO) glass via a metal-organic decomposition method (see "Methods", the photograph in Supplementary Fig. S12). The thickness of the $BiVO_4$ photoanode was approximately 200 nm (Supplementary Fig. S13). XRD measurements (Supplementary Fig. S14) were consistent with what has been previously observed in relevant studies[21,22]. Scanning electron microscopy (SEM) images (Fig. 4b) revealed the nanoporous morphology of the $BiVO_4$ photoanode, which can increase the contact area with seawater, improving charge transfer efficiency and reducing surface pH variations. The presence of Bi and V was confirmed through SEM element mapping (Fig. 4c and d). To accelerate oxygen evolution kinetics in seawater, NiFe(OH)$_x$ catalysts were deposited on the $BiVO_4$ photoanode surface by dip coating[23]. A thin $CrO_x$ layer was then photo-deposited onto the NiFe(OH)$_x$ catalysts to prevent $Cl^-$ oxidation and the subsequent

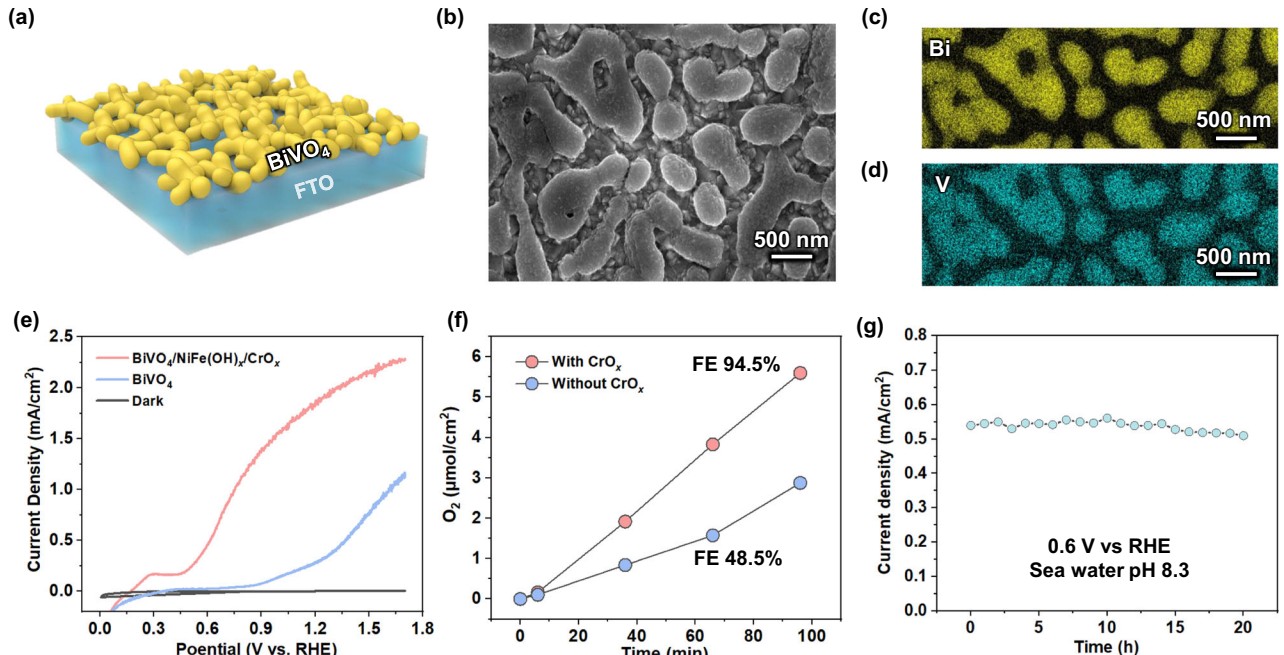

**Fig. 4 | BiVO₄ photoanodes for seawater oxidation. a** The schematics of BiVO₄ photoanodes. **b** The scanning electron microscopy image, and corresponding element mapping of Bi (**c**) and V (**d**). **e** J-E curves of BiVO₄ photoanode with or without catalysts under illumination and in the dark. **f** Time course of O₂ gas production. **g** Stability of BiVO₄ photoanode at 0.6 V vs RHE.

corrosion of the BiVO₄ photoanodes in seawater. The existence of CrOₓ on the BiVO₄ photoanode surface was confirmed by XPS (Supplementary Fig. S15).

The PEC oxygen evolution measurements were performed under AM 1.5 G illumination (100 mW/cm²) in seawater with a pH of 8.3 at ambient conditions. The BiVO₄ photoanode with NiFe(OH)ₓ/CrOₓ catalysts exhibited a current density of 1.8 mA/cm² higher than the 0.5 mA/cm² without cocatalysts at 0.6 vs RHE (Fig. 4e). The NiFe(OH)ₓ catalyst enhanced the kinetics of oxidation reaction, allowing for maximum ABPE of 0.51% (Supplementary Fig. S16). BiVO₄ photoanodes with CrOₓ exhibited the same onset potential and saturated current density (Fig. 4f), indicating that the thin CrOₓ layer does not impede photogenerated hole charge transfer. Under AM 1.5 G illumination, an O₂ Faradaic efficiency of 94.5% at 0.6 V vs RHE over a 2 h period was achieved, suggesting that the majority of the photogenerated holes were utilized for oxygen evolution, with significant oxygen bubble formation also observed (Supplementary Fig. S17). In addition, the BiVO₄ photoanode with NiFe(OH)ₓ catalysts exhibited a robust photocurrent density after 20 h (Fig. 4g). The color of the seawater remained clear after 20 h of operation (Supplementary Fig. S18). The generation of hypochlorite (ClO⁻) was negligible, therefore, as confirmed with hypochlorite detection. The high O₂ FE and the robust stability were attributed to the CrOₓ layer, which was reported as a Lewis acid to effectively modulate the local reaction microenvironment of the NiFe(OH)ₓ water-oxidation sites to effectively prevent chloride attack[24,25]. Therefore, protons (H⁺) can be continuously released during seawater oxidation at the BiVO₄ photoanode.

## pH map above photoelectrodes reveals H⁺ confinement to photoelectrode surfaces

We can couple acid-base reaction kinetics among the $HCO_3^-$, $CO_2$, $H^+$, and $OH^-$, with the shear flow of seawater to alter pH spatial distribution and generate excess $CO_2$(aq) in situ. Confocal fluorescence microscopy[26] and scanning laser microscopy[27] are promising techniques for pH mapping. Confocal fluorescence spectroscopy in a scanning confocal Raman microscope with a point-by-point scan mode was used because our in situ flow cell requires upward facing of BiVO₄ photoanodes to reflect the realistic photoreactor operating conditions (Fig. 5a). We took advantage of this customized tool to quantify and visualize pH (Fig. 5b), to eventually achieve indirect measures of $CO_2$ concentration, $CO_2$ flux, and $H^+$ flux, quantities that are otherwise difficult to measure (experimental detail shown in "Methods").

Spatial mapping revealed a uniform pH of ~ 8.3 across the entire BiVO₄ photoanode at open circuit potentials (Fig. 5c). This observation suggests that the introduction of flow confines the acidification process to the BiVO₄ photoanode surfaces that uniformly generate H⁺, whereas the pH of the seawater at >100 μm above the photoanode surface plane remains identical to the inflow seawater pH. Each pH data point was determined from the pH-indicator emission spectra, with an emission peak at 650 nm indicating a pH value of ~ 8.35 (Supplementary Fig. S19), which is consistent with the seawater's pH of 8.30 measured by a pH meter. We mapped spatial pH variations from 0–100 μm above the BiVO₄ photoanode surface during PEC seawater oxidation at a current density of 0.5 mA/cm². Under these conditions, the near-surface fluorescence spectra revealed an increase at the 580 nm emission peak, indicating a decrease in pH during seawater oxidation (Supplementary Fig. S20). At a flow velocity of 0 m/s, the pH is lowered to ~ 4.3 at 0–100 μm above the BiVO₄ photoanode surface (Fig. 5d), indicating that the water oxidation reaction at the BiVO₄ photoanode surface significantly alters the local pH. When the flow velocity is increased to 0.16 m/s, pH mapping revealed four distinct regions colored blue, green, yellow, and red. Near the BiVO₄ photoanode surface was the blue region, corresponding to lower pH and attributed to the higher proton concentration at the photoanode surface (Fig. 5e). At 60 μm above the photoanode surface, there was a gradual increase in the pH, indicated by the yellow region that continued until it approached the red region. As the flow velocity increased from 0.16 to 0.77 m/s, there was a corresponding expansion of the red area on the pH maps (Fig. 5e–h). This suggests that convective flow serves to effectively confine generated protons to the photoanode surface, to prevent their diffusion into the bulk seawater solution, and to enhance $CO_2$ extraction from seawater (Supplementary Fig. S21).

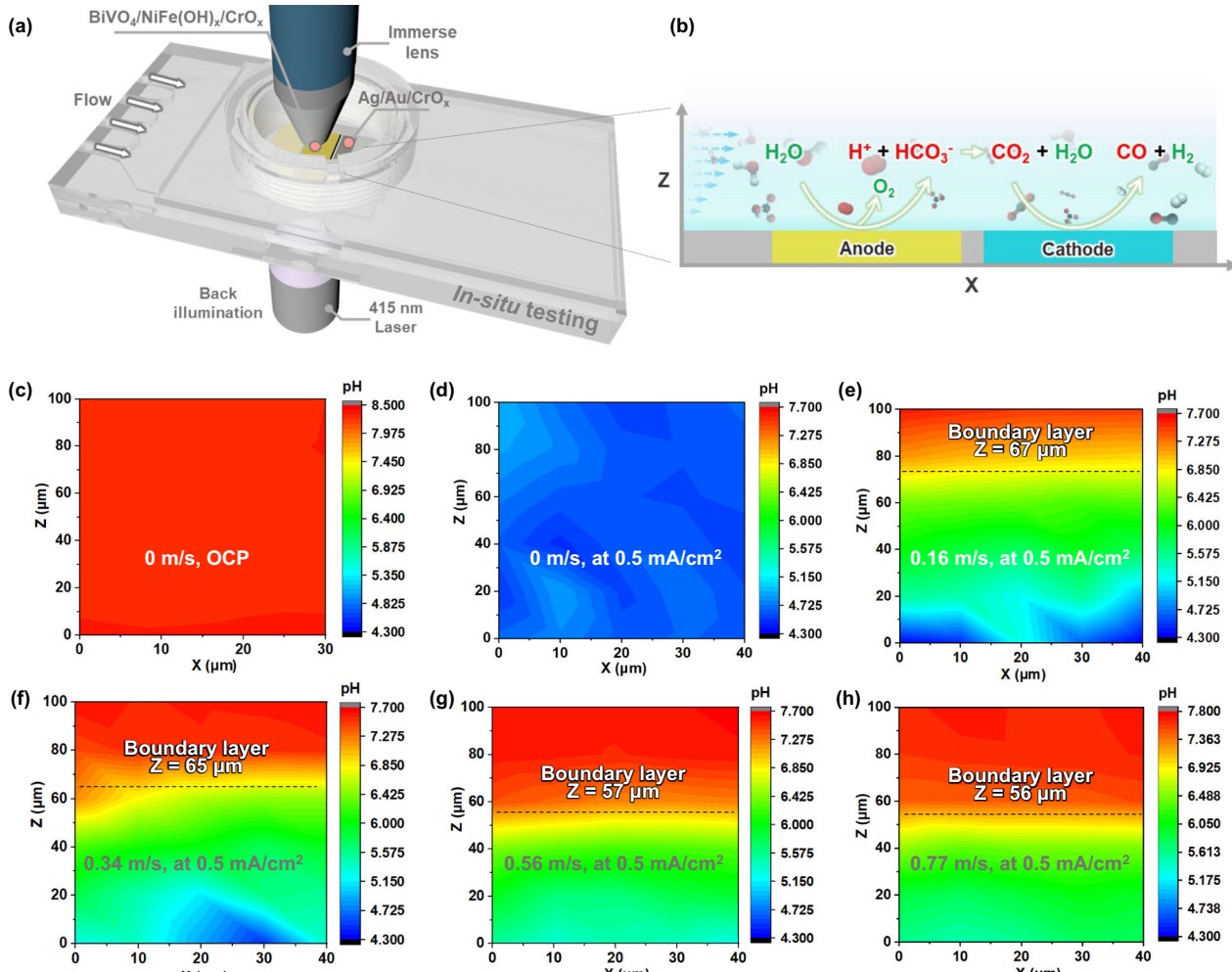

**Fig. 5 | In situ pH measurement on BiVO₄ photoanode for water oxidation.**
**a** The schematics of in situ scanning confocal fluorescence spectroscopy set-up for pH measurement. **b** The zoom-in illustration of water oxidation and $CO_2$ reduction reaction during the in situ measurement. The data was collected at $x = 0.8$ cm at the anode along the length of the anode. **c** pH profile on BiVO₄ photoanode under OCP. pH profiles on BiVO₄ photoanode at 0.5 mA/cm² under flow velocities of 0 (**d**), 0.16 (**e**), 0.34 (**f**), 0.56 (**g**), and 0.77 m/s (**h**). The $X$-axis represents the direction of the flow field, 0 μm is upstream, 40 μm is downstream. The $Z$-axis represents the distance from the BiVO₄ photoanode surface, 0 μm indicates the electrode surface and $Z = 100$ μm indicates 100 μm above the electrode surface. The dashed lines in (**e**−**h**) indicate respective boundary layer positions. A constant boundary layer thickness was calculated for each flow rate because the pH maps were measured near the end of photoanodes where these pH boundary layer profiles appear to be flat within only 30 μm.

To reveal the confinement effect of flow on H⁺ and $CO_2$ concentrations and transport flux, we calculated the boundary layer thickness under various flow velocities based on mass transport physics. We focus on the spatial distribution of the H⁺ concentration from the anode surface to the bulk solution. Such a pH map undergoes a smooth transition under the synergistic influence of convection and diffusion. This is known as the concentration boundary layer effect and manifests itself as a gradual change in the density of H⁺ and $H_3O^+$ ions from the electrode interface to the bulk solution. It is worth noting that although our discussion focuses on H⁺, this concentration boundary layer effect can be extended to other species present in the electrolyte including $CO_2$(aq), $HCO_3^-$, and OH⁻ (Supplementary Fig. S22). In this analytic calculation, for simplicity, we assume that H⁺ ions are generated uniformly on the anode surface and then diffuse into the bulk solution. An $x$-axis is established from the upstream to the downstream direction with the upstream edge of the anode as the origin, and a vertical $z$-axis is established upward from the anode surface (Fig. 6a). Drawing inspiration from Pohlhausen's integral approach to solve for boundary layer thickness, we assumed a linear concentration distribution for the H⁺ and $CO_2$ flux as a function of distance from the electrode surfaces[28]. Therefore, we propose a semi-quantitative Eq. (1)

for calculating the thickness of the H⁺ boundary layer, denoted as $\delta(x)$. A point is identified at $z = \delta$ where [H⁺] becomes exactly zero, demarcating the boundary between the region affected by the anode surface and the bulk solution. The derivation of Eq. (1) is shown in the Supplementary Materials (Page 2), where $u_O$, $k$, $D$, and $d$ are the average flow velocity in the reactor, the velocity gradient, the diffusion coefficient of H⁺, and the thickness of the reactor, respectively.

$$\delta \approx \left(\frac{6Dx_0}{k}\right)^{\frac{1}{3}} = \left(\frac{x_0 Dd}{u_0}\right)^{\frac{1}{3}} \quad (1)$$

At the same flow velocity ($u_O$), the downstream boundary layer thickness calculated by Eq. (1) is thicker than upstream (Fig. 6a). This can be attributed to higher downstream $CO_2$(aq) and proton concentrations. As the flow velocity increases, the boundary layer thickness decreases (Fig. 6a) because the convective mass transport flux in the $z$-direction increases with increasing velocity (Supplementary Fig. S23). At a fixed position in the $x$-axis direction (take $x_0 = 0.8$ cm as an example), the boundary layer thickness decreases from 77 μm at a flow velocity of 0.16 m/s to 40 μm at a flow velocity of 0.77 m/s

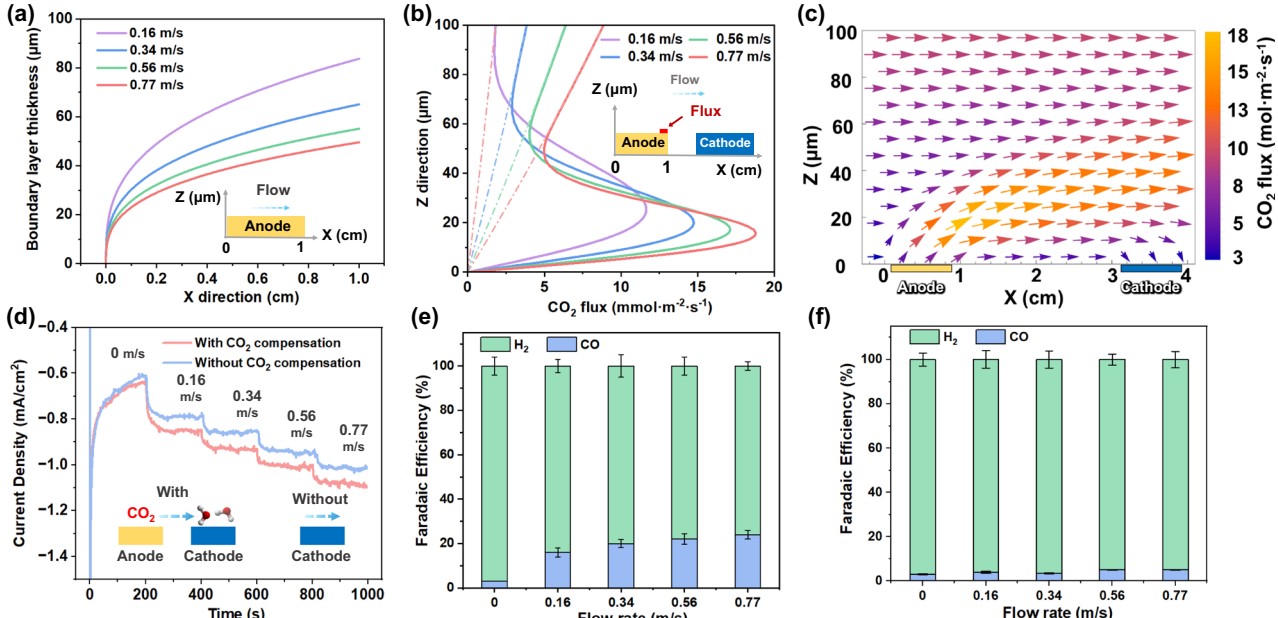

**Fig. 6 | Convective mass transfer flux calculation and experimental validation for CO₂ capture and conversion. a** The calculated boundary layer thickness of anode at different flow velocities, $u_0$ and flow distance $x_0$, labeled as *X*-axis. **b** convective flux of $CO_2$ at the end of anode ($X = 1$ cm) at different flow velocities. **c** vector flux profile of $CO_2$ at 0.77 m/s. **d** flow velocity-dependent current density with and without $CO_2$ compensation at −0.5 V vs RHE. **e** Faradaic efficiency of CO and $H_2$ at different flow velocities with $CO_2$ compensation. **f** Faradaic efficiency of

CO and $H_2$ at different flow velocities without $CO_2$ compensation. The current density applied in (**a–c**) was 0.5 mA/cm². The error bars in (**e**) and (**f**) represent the standard deviations of three independent measurements of the same samples. The voltage applied in (**d–f**) was − 0.5 V vs RHE. The *X*-axis represents the flow field's direction, where 0 cm is upstream, and 1 cm is downstream. The *Z*-axis represents the distance from the anode surface, where 0 μm is the electrode surface and 100 μm is above the electrode surface.

(Fig. 6a). It is particularly noteworthy that the thickness we obtained from Eq. (1) is consistent with the results of the pH map in our experiments (Fig. 5e–h), which validates the accuracy of the model. The comprehensive effects of boundary layer flow are to increase ionic conductivity, reduce pH gradient and $CO_2(aq)$ concentration overpotential losses, and keep the local pH close to neutral pH to improve the stability of the BiVO₄ photoanode[29]. This analysis supports the observed stable photocurrent of BiVO₄ photoanodes during the 20-hour stability test (Fig. 4g).

**Modeling time-dependent $CO_2(aq)$ convective-diffusive transport coupled with acid-base reactions of non-equilibrium generated H⁺ and OH⁻ with $CO_2(aq)$**

Although the pH mapping and the analytical boundary calculation indicated that the flow could alter the concentration distribution of H⁺ and $CO_2(aq)$ species, quantitative insights into their reaction-transport behaviors are still lacking. Thus, we quantitatively correlate the $CO_2(aq)$ molecular flux with the observed selectivity, i.e., $CO_2$-to-CO conversion partial current density. We then introduce chemical reactions into the fluid flow physics of our model. During $CO_2(aq)$ transport, both $CO_2(aq)$ and protons encounter and react with OH⁻, HCO₃⁻, and CO₃²⁻ ionic species (Fig. 5b). Their consumption competes against the $CO_2(aq)$ transport flux that promotes in situ $CO_2$ conversion. Therefore, we utilized COMSOL Multiphysics to elaborate on the impacts of flow on mass transfer and reaction selectivity.

Our COMSOL model considered the diffusion of all chemical species dissolved in seawater, while accounting for the real-time acid-base speciation reactions of all dissolved carbon species (Supplementary Fig. S24) (simulation details are shown in supplementary materials, page 43). The dashed line in Fig. 6b represents the linear distribution assumption for the $CO_2(aq)$ flux, which is consistent with the linear flux-position relationship that exists outside the boundary layer, according to our model calculation. The Fig. 6a plot allows us to compare the COMSOL modeling outcome to our analytical boundary

layer analysis: based on Eq. (1), the boundary layer thicknesses at $x = 0.8$ cm along the anode at flow velocities of 0.16, 0.34, 0.56, and 0.77 m/s are 77, 60, 51, and 45 μm, respectively. In comparison, the numerical boundary layer thickness at the same position and flow velocities are 80, 60, 50, and 42 μm, respectively. Taking the condition of flow velocity of 0.77 m/s as an example, the results obtained from the COMSOL modeling match well with those derived from the fluid mechanics analysis (Supplementary Fig. S25). Furthermore, the pH map contours quantitatively match the flow boundary (Fig. 5e–h), which is derived from the numerical calculations considering both acid-base reaction and convection-diffusion transport. These strong correlations under the multiple specific and varying parameters of flow and spatial positions not only indicate that the assumptions proposed during our numerical calculations are reasonable but also confirm the accuracy and applicability of in situ spectroscopy for monitoring local pH during PEC water splitting and CO₂R.

Across all flow velocities, the observations made from our pH mapping using in situ fluorescence measurements in a fluid flow (Fig. 5e–h) closely matched the trends of the COMSOL-simulated boundary layers (Supplementary Fig. S26). This validates our model and allows us to visualize $CO_2(aq)$ generation, $CO_2(aq)$ transport flux, and consumption behavior, along the photoanode-photocathode flow path. The cross-section located 1 cm from the anode's far end ($x = 1$) was taken to illustrate the distribution of $CO_2(aq)$ flux (Fig. 6b). At this specific cross-section, $CO_2(aq)$ flux peaks within a 40 μm range from the photoelectrode surface in the *z*-direction. The peaking of the $CO_2(aq)$ flux was observed at all flow velocities. As the flow velocity increased from 0.16 to 0.77 m/s, the $CO_2(aq)$ flux increased from 11.7 to 18.8 mmol·m⁻²·s⁻¹, respectively, demonstrating the ability of the flow field to enhance the $CO_2(aq)$ flux. Correspondingly, the respective CO selectivity was 3% and 21%. $CO_2(aq)$ generated through acidification was concentrated within a specific height range of e.g., 0–40 μm, closely matching what was determined with our semi-quantitative model. During $CO_2(aq)$ transport at flow velocities of 0.77 and 0.16 m/s from

the anode to the cathode, the $CO_2(aq)$ fluxes at the front end of the cathode were 13.53 and 7.5 mmol·m$^{-2}$·s$^{-1}$, respectively (Supplementary Fig. S27). This indicates that higher flow velocities correspond to higher $CO_2(aq)$ flux near the cathode surface, which is consistent with our findings that the $CO_2R$ selectivity for CO production and the STF efficiency increased at higher flow velocities (Fig. 3a). Moreover, the vector profile of $CO_2(aq)$ flux at 0.77 m/s provides an overview of $CO_2$ generation, transportation, and conversion (Fig. 6c). Although $CO_2$ diffuses upward, it is confined by the boundary layer flow and diffuses downward to the cathode surface. Therefore, a well-defined boundary layer flow can confine solution species near photoelectrodes to reduce the loss of electrochemically active reactants such as $CO_2(aq)$. The arrows indicate a downward $CO_2(aq)$ flux on the cathode surface, relevant to the Faradaic $CO_2$ reduction photocurrents at the Si photocathode surface, indicating the presence of diffusive mass transfer when $CO_2(aq)$ is transported to the cathode surface. This downward flux field also indicates the presence of diffusive mass transfer to the photocathode surface. $CO_2(aq)$ is constantly arriving at the Ag-Au/$CrO_x$ cocatalysts, adsorbing and reacting on the Au-Ag particle surfaces. The $CO_2(aq)$ consumption on the cathode surface lowers the local $CO_2$ concentration and sustains the downward $CO_2(aq)$ flux (other flow velocities are shown in Supplementary Fig. S28). The magnitude of $CO_2(aq)$ flux influences the rate of reaction on the cocatalyst active sites, as well as affects the $CO_2$ adsorption coverage and, thus, the observed $CO_2R$ selectivity.

## Chemical-species confinement under a boundary layer flow allows photoanodes to generate $CO_2(aq)$ flux effectively

By applying a seawater flow, the $H^+$ produced at the anode and the $CO_2(aq)$ generated from $H^+$ acidification can be confined within the boundary layer. We further used a model flat-plate flow reactor to quantify and benchmark the effect of $CO_2(aq)$ flux compensation on $CO_2R$ Faradaic efficiency. The flat-plate reactor design allows us to apply the validated multiphysics model to evaluate the rate and selectivity limit of seawater-based solar fuel production and guide further performance improvement.

An Ag-Au/$CrO_x$ cathode and an FTO/NiFe(OH)$_x$/$CrO_x$ anode were employed (fabrication details in the "Methods"), and the catalysts used are the same as photoelectrodes. To compare the current density and selectivity of cathodes with and without acidification effects from upstream anodes, the terminology used is as follows: "with $CO_2$ compensation" signifies that the anode was placed side-by-side with the cathode within the same reactor, while "without $CO_2$ compensation" indicates that the anode was situated outside the cell, separate from the cathode. The flow velocity-dependent chronoamperometry data reveals that under static conditions, both with and without $CO_2$ compensation, configurations exhibit comparable current densities (Fig. 6d). However, as the experiment progresses, the current densities steadily decline. This trend suggests an insufficient mass transfer of reactants, i.e., $H^+$ and $CO_2$, to adequately sustain the reductive current. The scenario takes a different turn when introducing the flow into the cell. In this case, the current densities remain stable and exhibit an increase with the ascending flow velocities within the range of 0 to 0.77 m/s, implying that the mass transfer is enhanced. The increasing current densities indicate improved STF conversion efficiency with increasing flow velocities, for PEC $CO_2$ reduction and broadly for water splitting and other fuel-forming photocatalysis. Notably, the reductive current densities observed with $CO_2$ compensation are greater than those observed without $CO_2$ compensation. This promotion in current can be attributed to the elevated $CO_2$ concentration on the cathode surface in our configuration. These observations also indicate that the flow-induced coupling between photocathodes and photoanodes is a unique feature in (photo-)electrochemical devices with a boundary layer flow design. The gas composition analysis indicates that under flow velocities of 0 m/s and 0.77 m/s, the faradaic efficiency for CO and

$H_2$ is 3% and 97%, respectively. Since in situ generated $CO_2$ can be consumed through acid-base reactions during its transport from the anode to the cathode, a lack of flow can result in the re-absorption of $CO_2$ back into the bulk seawater due to these reactions. In this case, photo-generated electrons primarily favor the reduction of $H^+$ to form $H_2$ under these conditions.

The CO Faradaic efficiency increases from 3% to 21% as the flow velocities increase, highlighting the crucial role played by mass transfer and $CO_2$ compensation for $CO_2$ in situ conversion (Fig. 6e). Our validated COMSOL simulations are extendable to the flat-bed design under hypothetical experimental conditions for parameter-bound analysis. Because flow transport and $CO_2R$ catalysis are in a tandem sequence, we introduce the analysis of the Damköhler number (Da) to evaluate the mass transfer of $CO_2(aq)$ in the flow PEC device, to help elucidate the role of $CO_2(aq)$ transport flux in its molecular flux catalysis. Da is defined as the ratio of the flow transport timescale from the anode to the cathode, to the chemical lifetime of in situ generated $CO_2(aq)$. For flow velocities of 0.16, 0.34, 0.56, and 0.77 m/s, the corresponding Da are 0.37, 0.22, 0.16, and 0.13, respectively. In this system, the Da numbers are significantly less than 1, indicating that the in situ $CO_2$ generated at the anode is transported to the cathode with minimal losses due to acid-base reaction consumption and out-diffusion to bulk seawater. Upon reaching the cathode surface, the $CO_2(aq)$ undergoes effective adsorption and $CO_2R$ catalysis, due to their fast millisecond timescale. This molecular flux catalysis proceeds the $CO_2(aq)$ flux transport process that is benchmarked by Da numbers. The partial current for CO production increases with the flow velocities, providing further evidence that precisely controlled seawater flow serves to accelerate both the overall reaction rate (current density) and enhance CO selectivity (Fig. 6f). It is noteworthy that comparable CO selectivity was observed within a range of flow velocities, provided the photocurrents were under the same order of magnitude. Da numbers of <1 under wide-ranging flow velocities confirm the feasibility of our system design, and further support the high CO selectivity and high carbon utilization efficiency.

In addition, reducing the distance L between the anode and cathode down to 0 cm can also decrease the Da, thereby enhancing the efficiency of $CO_2R$. We further conducted simulations to evaluate the maximum theoretical CO Faradaic efficiency, i.e., selectivity, for different distances between the anode and cathode. The limiting CO selectivity is expected to be 30% and 50% for photoanode-photocathode distances of 2 cm and 0 cm, respectively (Supplementary Figs. S29 and S30), by comparing the calculated $CO_2(aq)$-flux-limited photocurrent density with the experimental photocurrent density under the simulation conditions. The CO selectivity obtained in this work is approaching the 30% theoretical limit for generating CO gas from seawater. Convection flow will bring $CO_2(aq)$ molecule flux to the cathode surface under non-equilibrium conditions, thereby enhancing the reaction selectivity of $CO_2R$. The flat plate reactor (Fig. 6e and f) and vortex flow reactor (Fig. 3b–d) exhibit comparable $CO_2R$ performance, indicating that the well-defined vortex flow achieved by modeling-guided design and 3D printing can effectively enhance convective mass transport and confine the $CO_2(aq)$ molecular flux within the vortex flow.

The photocurrent density of the BiVO$_4$/Si photoelectrodes was measured to be 0.5 mA cm$^{-2}$ in flowing seawater of 2.3 mM HCO$_3^-$ under simulated 1-sun illumination. The improved photocurrent due to electrolyte flow indicates the reduced transport loss for PEC devices in general, setting a high 0.71% STF efficiency. The 0.5 mA cm$^{-2}$ photocurrent is used across in situ Fluorescence spectroscopy and COMSOL modeling throughout this study. We further showed that 70% CO selectivity and 90% carbon-product selectivity can be achieved via multi-pass photoelectrocatalytic reactions (Supplementary Fig. S31), indicating the upper limit for this technology to extract carbon products from seawater in the near future. Further improvement in STF

and CO selectivity requires combined experiment and modeling efforts, as we showed, to synergistically improve photocurrents per geometric illumination area, ensure sufficient $CO_2$(aq) flux, and reduce transport losses. We take $BiVO_4$/Si model photoelectrodes to show PEC $CO_2$R rates, with proper design, should not limited by $CO_2$(aq) transport flux under flowing seawater containing only 2.3 mM $HCO_3^-$. Our approach is distinctive to conventional PEC devices where STF efficiency improvement requires replenishing sufficient $CO_2$ reactants and enhancing light absorption, which, however, cannot address the low carbon efficiency issue.

This work reports $CO_2$(aq) flux catalysis enabled by a well-defined boundary layer flow for $CO_2$(aq) in situ generation from dissolved bicarbonates and subsequent conversion into fuels using seawater and sunlight as the only inputs. We demonstrate a regime in which the active $CO_2$(aq) reactant continuously flows to the Ag-Au cocatalysts supported on Si photocathodes, even when the local $CO_2$(aq) reactant concentration is near zero. The boundary layer flow facilitates an effective constraint of $CO_2$(aq) near the cathode surface as well as the convective-diffusion transport of $CO_2$(aq) within the boundary layer back to the cathode surface. As a result, the CO Faradaic efficiency exhibits a notable improvement from 3% to 21%, approaching the theoretical limit as the laminar flow velocity increases 0.77 m/s. With this PEC flow reactor design, we achieved a high $BiVO_4$-based STF efficiency of 0.71%. The reactor developed in this study can operate in 1000-times lower carbon concentration (Fig. 1b) than conventional PEC configuration (Fig. 1a), making seawater utilization feasible and practical.

Further performance improvements can benefit from the as-reported combined experimental and modeling approach to not just employ narrower bandgap photoanodes such as $Ta_3N_5$[30] but also to co-optimize $CO_2$(aq) flux in the dissolved carbon media with the rates of light-driven catalysis. The flow reactors can be repeated and assembled at scale, as shown in Supplementary Fig. S40, floating on seawater or trailing a boat. The product gas can be continuously collected from the reactor's headspace through the gas outlet and routed to a gas-collection tank for further compression and separation of syngas. These reactors directly demonstrate the potential for large-scale solar production of fuels and chemicals. If we separate $O_2$ through absorbents and combine membrane separation[31] with the flow photo-reactor in real-time, we can generate syngas, which can undergo hydrogenation reactions for liquid fuel production[32,33]. The 21% CO selectivity of our current PEC devices eventually produces $H_2$ and CO syngas at a 4:1 ratio in a single pass, which is within the idea range for subsequent thermocatalytic hydrogenation to make liquid fuels[34,35]. So far, we pump water to achieve the flow, but our reactor could float on the ocean and rely on the convection caused by ocean currents to enhance mass transfer between the $BiVO_4$ photoanode and Si photo-cathode to capture and convert $CO_2$. Alternatively, the reactor could be mounted on a ship and utilize its controlled flow field, replacing the peristaltic pump in this work. For dimensional scalability, the photo-reactor can also adopt long photoelectrode stripes, which are extended along the out-of-plane direction in Fig. 2b. This work establishes flow-based flux catalysis, which could be applied to various redox reactions, such as water splitting, hydrogen peroxide production, and methane selective oxidation, where elementary reaction steps, including proton reduction, selective oxygen reduction, and electrochemical C-H bond activation, are spatially separated.

## Methods

### Materials

Seawater salt was obtained from the Instant Ocean. Vanadyl acetylacetonate $(VO(acac)_2, 99.0\%)$ and Bismuth nitrate pentahydrate $(Bi(NO_3)_3 \cdot 5H_2O)$ are supplied by Millipore Sigma. Dimethyl sulfoxide (DMSO, AR) is obtained from Sigma-Aldrich. 18.25 MΩ cm ultra-pure water supplied by a Millipore system was used in the entire experimental process. All the reagents were used directly without further purification.

### In situ pH mapping measurement

To measure the pH spatial distribution in a seawater flow over the $BiVO_4$ photoanode surfaces, Carboxy SNARF-1 was selected as the fluorescent pH indicator due to its emission spectrum exhibiting a pH-dependent wavelength shift. pH at a given point was measured by calculating the fluorescence intensity at two different emission wavelengths (580 and 650 nm)[36,37]. $BiVO_4$ photoanode can operate in both back- or front-illumination (Supplementary Fig. S32). To mitigate the potential influence of front-illumination (415 nm laser) on the pH indicator dye's peak intensities and position, back-illumination was employed (photographs of the apparatus used in the pH measurement are shown in Supplementary Figs. S33 and S34) The 532 nm laser is used to excite pH indicator, and 415 nm laser provides uniform illumination for $BiVO_4$ photoanodes. The pH indicator was first calibrated using various standard solutions. An immersion water lens (60 x) was used for high-resolution spectra acquisition, and the Raman laser was focused on the top surface of $BiVO_4$ photoanodes (Fig. 5b and Supplementary Fig. S35). The COMSOL simulation shows that the pH reaches a steady state within 1 s (Supplementary Fig. S36). During experiments, we ensure the reactions run long enough (5 min) to reach this equilibrium before conducting pH measurements.

### Characterization

The thickness of the a-Si on the polished Si (100) monitor substrate was obtained by a spectroscopic ellipsometer (M-2000 DI, J. A. Woollam Co., Inc.). The morphology and microstructure of the samples are characterized by field emission scanning electron microscope (FE-SEM, Hitachi S-8100, 5 kV). The UV-vis spectra are obtained on a SHIMADZU UV-2550 spectrophotometer. XPS measurements were performed using a PHI Versa Probe II Scanning XPS Microprobe equipped with a monochromatic Al source. The crystal structures were confirmed by XRD 271 using a Rigaku SmartLab X-ray Diffractometer in a grazing incident mode. Fluorescence spectra were collected using a confocal Raman microscope (LabRAM HR Evolution, Horiba Jobin Yvon). The excitation source was a 532 nm laser. A 60X water-immersion objective (LUMPlanFL, Olympus) was used. For the in situ pH measurement, the Blue LED light with 415 nm single-wavelength light excitation (Item# M475L4h, FWHM = 17 nm, Thorlabs) was used as the light source for the $BiVO_4$ photoanode.

### PEC measurements

PEC measurements were carried out using a three-electrode config-uration with the prepared electrode as the working electrode, saturated Ag/AgCl as the reference electrode, and carbon as the counter electrode. The potentials obtained from each measurement were converted into values against the reversible hydrogen electrode using the Nernst equation: $E_{RHE}(V) = E_{Ag/AgCl} + 0.059 \times pH + 0.197$. An SP-300 Biologic potentiostat was used to control the potentials and record the data (without iR correction). The J-V curves of the samples were measured with a scan rate of 50 mV s$^{-1}$ under irradiation with a 300 W Xenon lamp (Newport) equipped with an AM 1.5 G filter. Light-driven $CO_2$ reduction on the photocathode was performed under AM 1.5 G illumination (100 mW/cm$^2$) in seawater with $CO_2$ purge (pH 7) under ambient conditions. The gas products were analyzed by a gas chromatograph (Shimadzu, GC-8A) equipped with a thermal conductivity detector. Ar was used as a carrier gas. To quantify the flow velocity, the volumetric flow rate was divided by the cross-sectional area of the tubing.

### Chlorine detection

After the stability test, the addition of 15 mL of 0.5 M KI (potassium iodide) to 10 mL of seawater would result in a color change if

hypochlorite ions ($ClO^-$) are present. The reaction between $ClO^-$ and iodide ions ($I^-$) would lead to the following color-indicative reaction: $ClO^- + 2I^- + 2H^+ = I_2 + Cl^- + H_2O$.

## Fabrication of BiVO$_4$ photoanodes

BiVO$_4$ photoanodes were synthesized by metal organic decomposition methods, Bi and V precursors with 1 M dissolved DMSO simultaneously. Then, the precursor solution was filtered by a 0.45 μm pore size filter. After that, the precursor solution is deposited onto FTO by spin coating at 1000 rpm for 20 s followed by 3000 rpm for 40 s, calcined in a tube furnace at 500 °C (ramping rate of 5 °C/min) for 2 h in the air. NiFe(OH)$_x$ catalyst with conformal coverage was deposited by a citrate-additive-assisted method.

## Fabrication of BiVO$_4$/Si tandem devices

The BiVO$_4$ photoanode is 2 cm in length and 1 cm in width, while the Si photocathode is 0.5 cm in length and 0.5 cm in width. The Indium was used to solder the back contact of the BiVO$_4$ photoanode together with the Si photocathode to allow the charges to transfer to the ohmic back contact. Then, it was further encapsulated by epoxy.

## Products collection and measurement

Collect the gas using a syringe through the rubber plug in the reactor (Supplementary Fig. S1a). Before the reaction, the reactor with Si/BiVO$_4$ electrode was immersed in simulated seawater without headspace. The headspace samples were analyzed by an SRI gas chromatograph (SRI 8610 C #3) by syringe injection. Unless specified, the injection volume was 1 mL. The GC is equipped with TCD (to quantify H$_2$, O$_2$, and N$_2$) and methanizer-equipped FID Detector (to quantify CO and CO$_2$).

## Data availability

All data generated or analyzed during this study are included in the published article and its Supplementary Information. Source data are provided in this paper.

## Code availability

All codes used in this study are provided upon request.

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

## Acknowledgements

S.H. gratefully acknowledges the financial support provided by the Division of Chemical Sciences, Geosciences and Biosciences, Office of Basic Energy Sciences, of the U.S. Department of Energy through an Early Career grant No. DE-SC0021953 for the photoelectrode development and photoelectrochemical interface characterizations. The authors thank Yuanzuo Gao (Yale University) for supporting the 3D printing efforts and Emily Q. Wang for improving device performance and data presentation. This research was supported in part by the Yale Planetary Solutions Project.

## Author contributions

S. H. supervised the project. S.H. and B.L. conceptualized the project. B.L., Z.Q., X.S., and H.S. developed the concept and carried out the experiments and data analysis. Catalyst synthesis procedures were developed and performed by W.Z. Y.Z., C.H., A.K., and R.Y. contributed to the development of the photoelectrode fabrication method. All authors discussed the results and participated in the writing of the manuscript.

## Competing interests

The authors declare no competing interests.
