## [Transparent Peer Review file · Nature Communications]

Solar-driven selective conversion of millimolar dissolved carbon to fuels with CO₂(aq) molecular flux generation

Corresponding Author: Dr Shu Hu

Version 0:

Reviewer comments:

Reviewer #1

(Remarks to the Author)

The authors have carefully addressed all my comments and have added substantive clarifications to the manuscript.

Although I did not ask for this in my first review, I would recommend adding a little further discussion on the following two aspects:

- Some suggestion for how the generated syngas can be practically captured while the device floats around on seawater / trails a boat. I realise this is a bit outside the scope of the study, but I think additional discussion would help to dispel concerns about the practicability of the device.

- The setting of the tilt angle of the photoanode/photocathode monoliths to 45 degrees. What range of tilt angles would still enable the desired flow in the device? The way I understand it, the penetration of light would rapidly decrease with depth below the water surface, meaning that scale-up of the photoabsorbers would be limited depth-wise. However, perhaps they could simply be scaled in length (i.e. in a direction parallel to the surface of the water, rather than perpendicular to it). In any case, a few words about the dimensional scalability (size constraint) would be a useful addition to the manuscript.

Reviewer #2

(Remarks to the Author)

I believe all the issues raised have been adequately addressed, and the manuscript warrants publication.

Reviewer #3

(Remarks to the Author)

In this manuscript, the authors proposed an interesting design of electrode array configuration to achieve efficient in situ generation of CO₂(aq) from mM-level HCO₃⁻, so that external bubbling of CO₂(gas), an obstacle for the practical use of large scale CO₂ reduction, was avoided. An optimized boundary shear flow was able to transport HCO₃⁻ from alkaline anode to acidic cathode to spontaneously produce CO₂(aq). In situ pH mapping and COMSOL simulation were employed to demonstrate the existence and the effect of such boundary shear flows. A CO selectivity of 20% as well as a STF efficiency of 0.71% was realized.

The general impression from this reviewer was that the study was nicely conducted and presented. Major conclusions were well supported. This reviewer thus supports the publication of this manuscript in Nature Communications if the authors could address the following technical concerns, which were mostly relevant to the experimental section on pH mapping upon the Editor's advice.

(1) In the present study, in situ pH mapping was enabled by a modified confocal-Raman microscope, in which the illumination was achieved with an inverted 415-nm laser. It was unusual (or even strange) to this reviewer.

1a. First, it would be natural to use a confocal FL microscope rather than confocal Raman to conduct the experiments. Please explain. The uses of confocal FL microscope in mapping 3-dimensional (particularly vertical dimension) pH distribution have been quite common and easy to handle. There are many relevant publications available (for instance. The Journal of Physical Chemistry Letters 2020 11 (17), 7042-7048; DOI: 10.1021/acs.jpcllett.0c01575). Because of many technical advantages, this reviewer strongly recommends to use confocal FL microscope, rather than confocal Raman, in this study.

1b. Second, in case if the authors do not have convenient access to confocal FL microscopes and they tended to use confocal Raman, it would be again natural to directly use the top-illumination laser equipped in confocal Raman. According to the authors' statement: 'To mitigate the potential influence of illumination (415 nm laser) on the pH indicator dye's peak intensities and position, back-illumination was employed. Please explain in detail on the 'potential influence'. This reviewer's point is that, any potential influence caused by a 415-nm top-illumination should also be there in back-illumination, unless the illumination quality (for instance, power density) was compromised. Please note that the vertical resolution was enabled by not only the emission pinhole during the collection of fluorescence photons, but also the excitation pinhole in the illumination path.

1c. In case if the authors would argue that the confocal Raman they had was not equipped with a top-illumination 415-nm laser, suitable solution should be using a different pH-sensitive dye that was compatible with the excitation wavelength. There are many kinds of commercially available pH-sensitive dyes with various excitation wavelength.

1d. In short, there are routine equipments and solutions to map the vertical distribution of pH via confocal FL microscope. The authors employed an unusual, and more importantly, imperfect approach to achieve this goal. At the same time, this reviewer agrees that such flaw can be fixed and may not damage the major value of this study after suitable improvements.

(2) The pseudo-color maps in Fig. 5 could be misleading.

2.a Because it is well-awared that the illutration was greatly manipulated by the color-map, it is necessary to display a real-curve of pH as a function of vertical distance for each flow rate (similar to Fig. S25), so that the boundary height (layer thickness) can be better justified.

2.b pH is a logarithm of proton concentration. One then wonders which curve is more meaningful here to determine the boundary: the pH-Z profile or the [H⁺]-Z profile? Please comment on this.

(3) Time-dependent pH map should be provided to understand the dynamic evolution of vertical boundaries. Is it temporally evolving or rather stable?

Version 1:

Reviewer comments:

Reviewer #1

(Remarks to the Author)

The authors have carefully considered all my queries and have responded in full. Therefore, I am happy to recommend publication of this manuscript.

(Remarks on code availability)

Reviewer #3

(Remarks to the Author)

This reviewer is satisfied with the revision and has no further questions.

(Remarks on code availability)

Point-by-point replies to the reviewer comments

Response to Comments of Reviewer #1:

General Comments R1: *The authors have carefully addressed all my comments and have added substantive clarifications to the manuscript.*

Although I did not ask for this in my first review, I would recommend adding a little further discussion on the following two aspects:

Response: We appreciate the reviewer's positive and constructive comments. Those additional questions and suggestions raised are very important and helpful, which make us think deeper, thereby improving the quality of the presentation and communication in this revision. We have addressed the reviewer's concerns by performing additional experiment tests, analysis, and modeling. Our responses to the reviewer's comment have been presented below.

Specific Comments R1-1: *Some suggestion for how the generated syngas can be practically captured while the device floats around on seawater / trails a boat. I realise this is a bit outside the scope of the study, but I think additional discussion would help to dispel concerns about the practicability of the device.*

Response: We appreciate the reviewer's insightful and valuable suggestions. This is an excellent point. To follow the reviewer's suggestion, we have added the related discussion in the revised manuscript Paragraph 2 Page 25 "The flow reactors can be repeated and assembled at scale, as shown in Fig. S40, floating on seawater or trailing a boat. The product gas can be continuously collected from the reactor's headspace, through the gas outlet, and routed to a gas-collection tank for further compression and separation of syngas. These reactors directly demonstrate the potential for large-scale solar production of fuels and chemicals." A mobile product-gas container, which is connected to the outlet of our floating photoreactors, can help to address the need for gas collection and further perform catalytic reforming for liquid fuels.

Fig. S40 Schematic diagram of gas collection for large-scale flow reactors floating on seawater.

Specific Comments R1-2: *The setting of the tilt angle of the photoanode/photocathode monoliths to 45 degrees. What range of tilt angles would still enable the desired flow in the device? The way I understand it, the penetration of light would rapidly decrease with depth below the water surface, meaning that scale-up of the photoabsorbers would be limited depth-wise. However, perhaps they could simply be scaled in length (i.e. in a direction parallel to the surface of the water, rather than perpendicular to it). In any case, a few words about the dimensional scalability (size constraint) would be a useful addition to the manuscript.*

Response: We acknowledge the helpful suggestion from the reviewer. To evaluate the effect of tilt angles on flow fields, we have conducted additional simulations for representative tile angles of 30° and 60°, to cover wide-ranging latitudes for photoreactor operation. The simulations show that when the tilt angles are 30° and 60°, the width and spacing of photoelectrode pairs are adjusted accordingly to maximize light capture. Therefore, we can adjust the tilt angle of the photoelectrodes based on the latitude of different regions to achieve maximum light utilization efficiency.

Fig. S3 The flow field simulations for representative tilt angles of 30° (a) and 60° (b).

To follow the reviewer’s suggestion, we have added the related discussion in the revised manuscript Paragraph 1 Page 9 “When the tilt angles are 30° (Fig. S3a) and 60° (Fig. S3b), respectively, their flow fields and velocity are similar to those observed at 45° (Fig. 3), indicating that the well-defined flow can bring in-situ generated CO₂(aq) from photoanode to photocathode. In principle, these angles can be adjusted to the latitude of reactor deployment, by design, to maximize light capture.”.

As for dimensional scalability, we do agree with the reviewer that the photoelectrodes require longer electrodes such as strips to avoid light absorption losses. To follow the reviewer’s suggestion, we have added the related description in the revised manuscript Paragraph 1 Page 26 “For dimensional scalability, the photoreactor can also adopt long photoelectrode stripes, which are extended along the out-of-plane direction in Fig. 2b.”.

Response to Comments of Reviewer #2:

General Comments R2: *I believe all the issues raised have been adequately addressed, and the manuscript warrants publication.:*

Response: We greatly appreciate the reviewer's encouraging and insightful comments. The questions and suggestions raised are both important and helpful, prompting us to continue thinking critically and ultimately improving the quality of our work.

Response to Comments of Reviewer #3:

General Comments R3: *In this manuscript, the authors proposed an interesting design of electrode array configuration to achieve efficient in situ generation of CO₂(aq) from mM-level HCO₃⁻, so that external bubbling of CO₂(gas), an obstacle for the practical use of large scale CO₂ reduction, was avoided. An optimized boundary shear flow was able to transport HCO₃⁻ from alkaline anode to acidic cathode to spontaneously produce CO₂(aq). In situ pH mapping and COMSOL simulation were employed to demonstrate the existence and the effect of such boundary shear flows. A CO selectivity of 20% as well as a STF efficiency of 0.71% was realized.*

The general impression from this reviewer was that the study was nicely conducted and presented. Major conclusions were well supported. This reviewer thus supports the publication of this manuscript in Nature Communications if the authors could address the following technical concerns, which were mostly relevant to the experimental section on pH mapping upon the Editor ' s advice.

Response: We thank the reviewer for the positive and encouraging note. We also thank the reviewer for their critical and thoughtful comments that have helped us clarify this unusual method used in our manuscript. We initiated this direction of employing confocal fluorescent (FL) microscopy to characterize *photoelectrochemical cells*. All technical concerns raised should *not* undermine our measurements and conclusions but rather show the need for clarification, due to our unconventional set-up: the *in situ* photoelectrochemical cells required both electrode illumination and spectral excitation. We are confident that our developed FL spectroscopic scan is suitable at this stage. We have done extra leg work to update the text from Raman to FL microscopy consistently using our responses to the reviewer's comments that have been presented below.

Specific Comments R3-1: *In the present study, in situ pH mapping was enabled by a modified confocal-Raman microscope, in which the illumination was achieved with an inverted 415-nm laser. It was unusual (or even strange) to this reviewer.*

1a. First, it would be natural to use a confocal FL microscope rather than confocal Raman to conduct the experiments. Please explain. The uses of confocal FL microscope in mapping 3-dimensional (particularly vertical dimension) pH distribution have been quite common and easy to handle. There are many relevant publications available (for instance. The Journal of Physical Chemistry Letters 2020 11 (17), 7042-7048; DOI: 10.1021/acs.jpcllett.0c01575). Because of many technical advantages, this reviewer strongly recommends to use confocal FL microscope, rather than confocal Raman, in this study.

Response: We thank the reviewer for the helpful comment and for reminding us of the norm of the field. We tried conventional scanning fluorescent (FL) microscopy before and fully understood the reviewer's concerns. We apologize for not thoughtfully explaining that and clearly describing the operational mode of the apparatus during the

in-situ fluorescence microscopy. We would like to clarify that unlike using fluorescent microscopy for electrochemical cells in the dark, PEC experiments require light illumination during pH testing. The front laser used for pH dye emission was 532 nm and the inverted 415 nm laser was used to illuminate the BiVO₄ photoanode area to generate electron-holes for in situ photoelectrochemical (PEC) measurement (please see the following schematic). Therefore, given these experimental and equipment constraints, we chose to use a confocal Raman microscope in its confocal fluorescent spectroscopy mode and perform a point-by-point scan.

Figure S33 (a) Schematic of customized flow device apparatus used for the in-situ fluorescence measurement in a fluid flow

To make it clear to readers, we have added the related description in revised manuscript “Confocal fluorescence microscopy and scanning laser microscopy are promising techniques for pH mapping. Confocal fluorescence spectroscopy in a scanning confocal Raman microscope with a point-by-point scan mode was used because our *in situ* flow cell requires upward facing of BiVO₄ photoanodes to reflect the realistic photoreactor operating conditions (Fig. 5a)” The reason for choosing a confocal Raman microscope to do confocal FL spectroscopy is that: In standard FL microscopes, laser excitation typically illuminates from below the sample. If the photoelectrode surface faces downward, gases like H₂, CO, and O₂ produced on the photoelectrode surface would desorb easily due to buoyancy, causing bubbles to cover the photoelectrode surface that could separate the electrode surface from the electrolyte. This would reduce mass transfer, slow the reaction rate, and affect product selectivity. To maintain realistic operating conditions, where sunlight hits the photoelectrode surface from above and buoyancy aids in gas desorption, we place the electrode facing upward in a confocal Raman microscope, which has laser excitation from above and allows us to collect FL spectral and intensity data.

We have cited the relevant references as well. We also added Figure S33a in the revised supplementary information.

Added references “

26. Pande, N. et al. Electrochemically Induced pH Change: Time-Resolved Confocal Fluorescence Microscopy Measurements and Comparison with Numerical Model. *J.*

Phys. Chem. Lett. **11**, 7042-7048 (2020)

27. Hicks, M. H. et al. Electrochemical CO₂ Reduction in Acidic Electrolytes: Spectroscopic Evidence for Local pH Gradients. *J. Am. Chem. Soc.* **146**, 25282-25289 (2024)

Specific Comments R3-2: *1b. Second, in case if the authors do not have convenient access to confocal FL microscopes and they tended to use confocal Raman, it would be again natural to directly use the top-illumination laser equipped in confocal Raman. According to the authors' statement: 'To mitigate the potential influence of illumination (415 nm laser) on the pH indicator dye's peak intensities and position, back-illumination was employed. Please explain in detail on the 'potential influence'. This reviewer's point is that, any potential influence caused by a 415-nm top-illumination should also be there in back-illumination, unless the illumination quality (for instance, power density) was compromised. Please note that the vertical resolution was enabled by not only the emission pinhole during the collection of fluorescence photons, but also the excitation pinhole in the illumination path.*

Response: We are very grateful for the reviewer's comment to help improve our manuscript clarity. To maintain realistic operating conditions, confocal Raman FL was selected, which allows the BiVO₄ photoanode to be placed facing upwards. As stated in Response R3-1, we apologize for not giving clear information of the apparatus used for the in-situ fluorescence in our work.

We mentioned the "potential influence" because we are worried that in the case of front-illumination, the 415 nm laser would affect the light excitation for fluorescent pH indicator dyes and thus the spectral collection, therefore we chose back-illumination. In this case, using external back-illumination does not compromise accuracy compared to the more conventional top-illumination through the measurement telescope, because most 415 nm laser can be absorbed or reflected by BiVO₄ photoanode.

Fig. S31 Schematic of back- and front-illumination configuration.

We employed the top-illumination in confocal-Raman, with 532 nm laser. 415 nm laser was used as an external light source to illuminate the BiVO₄ photoanode. BiVO₄ can operate in back- or front-illumination (Science, 2014, 343, 990; Nature Energy,

2018, 3, 53), please see the following figure. If the light source and the BiVO₄ light absorber are on different sides, it is called back-illumination. On the contrary, if the light source and the BiVO₄ light absorber are on the same side, it is called front illumination.

To make it clear to readers, we have added related descriptions in the manuscript Paragraph 3 Page 26 “BiVO₄ photoanode can operate in both back- or front-illumination (Fig. S31). To mitigate the potential influence of front-illumination (415 nm laser) on the pH indicator dye’s peak intensities and position, back-illumination was employed. The 532 nm laser is used to excite pH indicator, and 415 nm laser provides uniform illumination for BiVO₄ photoanodes.” We also added Figure S31 to the revised supplementary information.

Specific Comments R3-3: 1c. In case if the authors would argue that the confocal Raman they had was not equipped with a top-illumination 415-nm laser, suitable solution should be using a different pH-sensitive dye that was compatible with the excitation wavelength. There are many kinds of commercially available pH-sensitive dyes with various excitation wavelength.

Response: We appreciate the reviewer’s comments and suggestions. As mentioned in R3-2, 415 laser was for uniform electrode illumination; whereas 532 laser was for inline illumination through the confocal FL optics. We apologize again for our unclear schematic. We did the confocal FL spectroscopy and we have added a related description and schematic to point out our configuration in the revised manuscript.

To make it clear to readers, we have added related descriptions in the manuscript Paragraph 3 Page 26 “BiVO₄ photoanode can operate in both back- or front-illumination (Fig. S31). To mitigate the potential influence of front-illumination (415 nm laser) on the pH indicator dye’s peak intensities and position, back-illumination was employed. The 532 nm laser is used to excite pH indicator, and 415 nm laser provides uniform illumination for BiVO₄ photoanodes.” We also added Figure S31 to the revised supplementary information.

Specific Comments R3-4: 1d. In short, there are routine equipments and solutions to map the vertical distribution of pH via confocal FL microscope. The authors employed an unusual, and more importantly, imperfect approach to achieve this goal. At the same time, this reviewer agrees that such flaw can be fixed and may not damage the major value of this study after suitable improvements.

Response: We thank the reviewer for the encouragement and significant suggestion. As suggested by the reviewer, we have clarified the configuration we used and added the related description in revised manuscript. Although pH distribution mapping is commonly done by monitoring dye intensity changes at different pH levels, we could not follow this conventional approach due to the influence of reflective electrodes on the measured intensity. While we recognize that our method is not without limitations, we believe it represents the best compromise given the unique challenges of our

measurement conditions. The pH we measured with confocal-Raman is in good agreement with the simulation and pH meter test results (Fig. S20), so we have confidence in our approach.

Besides, it is important to note that the conventional confocal FL microscopy only allows us to track two bands instead of the whole spectra. In this early stage of method development, we are not confident that this conventional ratiometric approach is accurate enough for pH measurement, whereas spectral peak fitting is proven to yield accurate pH measurements.

Specifically, in our experiment, we measured the emission spectrum of SNARF-1. Given that SNARF-1 exhibits two distinct emission bands, this method minimizes interference from other spectral bands (e.g., shine light from the electrode surface), thereby enhancing the accuracy of pH measurement. This setup necessitates spectrum acquisition within the emission range of SNARF-1.

While other confocal fluorescence microscopes, such as spinning disk confocal microscopes, offer higher temporal resolution, our method provides improved measurement accuracy. Furthermore, these microscopes are not directly applicable to our experiment due to their incompatibility with spectral measurements, which are essential for our analysis.

To make it clear to readers, we have added the related description in the revised manuscript “Confocal fluorescence microscopy²⁶ and scanning laser microscopy²⁷ are promising techniques for pH mapping. Confocal fluorescence spectroscopy in a scanning confocal Raman microscope with a point-by-point scan mode was used because our *in situ* flow cell requires upward facing of BiVO₄ photoanodes to reflect the realistic photoreactor operating conditions (Fig. 5a).” We cited the important references below.

Added references “

26. Pande, N. et al. Electrochemically Induced pH Change: Time-Resolved Confocal Fluorescence Microscopy Measurements and Comparison with Numerical Model. *J. Phys. Chem. Lett.* **11**, 7042-7048 (2020)

27. Hicks, M. H. et al. Electrochemical CO₂ Reduction in Acidic Electrolytes: Spectroscopic Evidence for Local pH Gradients. *J. Am. Chem. Soc.* **146**, 25282-25289 (2024)

Specific Comments R3-5: (2) The pseudo-color maps in Fig. 5 could be misleading.

2.a Because it is well-aware that the illustration was greatly manipulated by the color-map, it is necessary to display a real-curve of pH as a function of vertical distance for each flow rate (similar to Fig. S25), so that the boundary height (layer thickness) can be better justified.

Response: We thank the reviewer for their valuable and significant comments. We have updated Figure 5 as shown below.

Fig. 5 In situ pH measurement on BiVO₄ photoanode for water oxidation. a, The schematics of in situ scanning confocal fluorescence spectroscopy set-up for pH measurement. b, The zoom-in illustration of water oxidation and CO₂ reduction reaction during the *in-situ* measurement. The data was collected at $x = 0.8$ cm at the anode along the length of the anode. c, pH profile on BiVO₄ photoanode under OCP. pH profiles on BiVO₄ photoanode at 0.5 mA/cm² under flow velocities of 0 (d), 0.16 (e), 0.34 (f), 0.56 (g), and 0.77 m/s (h). The X-axis represents the direction of the flow field, 0 μm is upstream, 40 μm is downstream. The Z-axis represents the distance from the BiVO₄ photoanode surface, 0 μm indicates the electrode surface, and Z = 100 μm indicates 100 μm above the electrode surface. The dashed lines in (e)-(h) indicate respective boundary layer positions. A constant boundary layer thickness was calculated for each flow rate because the pH maps were measured near the end of photoanodes where these pH boundary layer profiles appear to be flat within only 30 μm.

Considering that our measurements were conducted near the end of the BiVO₄ anode and that the measurements were only restricted to a 50 μm range in the x-direction where the boundary layer thickness for H⁺ concentration varies minimally with x within a measured range of only 30 μm, both the semi-quantitative analysis (Fig. 6a) and simulation results (Fig. S26) support this observation. Therefore, we believe that within the area shown on the pH map, the boundary layer thickness remains nearly constant or appears to be flat. To reduce measurement errors, we averaged the x-axis data and calculated a constant boundary layer thickness at the same flow rate.

The measured boundary layer thicknesses agree with the trend shown in Fig. S26a.

To accurately reflect the trend in boundary height changes, we intentionally normalized the color map to use the same color scale within the pH range of 4.3 to 7.7, except for the pH profile on the BiVO₄ photoanode under OCP, where the pH reaches 8.3 beyond the range of the other graphs. We chose a color map because it effectively displays both the X and Z dependencies. As for pH as a function of vertical distance, it can be easily interpreted by observing color changes along the Z-axis. We have made every effort to present the data as accurately and impartially as possible.

At flow rates of 0.16, 0.34, 0.56, and 0.77 m/s, the boundary layer thicknesses were 67, 65, 57, and 56 μm , respectively. It can be observed that the H⁺ boundary layer thickness decreases as the flow rate increases, showing a similar trend as in Fig. 26a. These experimental boundary layer thicknesses are consistently higher than COMSOL simulations possibly because the pH measurements at BiVO₄ photoanode surfaces could be overestimated due to reflection and other optical artifacts (sometimes measured pH > 5 compared to simulated values of 4.1).

To follow the reviewer’s suggestion and make it clear to readers, we will upload the raw data curves of pH to the online Dataverse repository as indicated in the “Data Availability” section.

Specific Comments R3-6: pH is a logarithm of proton concentration. One then wonders which curve is more meaningful here to determine the boundary: the pH-Z profile or the [H⁺]-Z profile? Please comment on this.

Response: We thank the reviewer for their valuable and significant comments. We present the results using the pH-Z graph. In these regions, [H⁺] is already close to zero, so pH is more meaningful to illustrate the variation of H⁺ concentration with Z.

Fig. R1 Simulated concentration profiles of H⁺ at x = 0.8 cm under 0.77m/s.

The profiles that the reviewer mentioned are about the boundary layer curvature at constant pH or [H⁺], where the boundary layer shape for pH and for [H⁺] appear to be

the same. pH provides a more intuitive representation and more accurate measure of the transition in $[H^+]$ near the boundary layer. Furthermore, pH is more relevant to experimental measurements. Here, we compare the $[H^+]$ -z and pH-z graphs based on COMSOL results, which have a higher spatial resolution. In the area highlighted by the red box (indicating the area near the boundary layer), the $[H^+]$ -z relationship no longer effectively captures the trend, whereas the pH-z relationship does.

Using the pH-Z graph, we can obtain the boundary layer thickness defined in the same way as in the $[H^+]$ -Z graph, but with a wide dynamic range to fit with the transport equations. Based on the same formula used in the COMSOL simulations to define boundary layer ($\frac{[i]_z - [i]_{\text{bulk}}}{[i]_{\text{surface}} - [i]_{\text{bulk}}} \times 100\% = 1\%$), and considering $[H^+]_{\text{bulk}} = [H^+]_{z=100\mu\text{m}}$, and $[H^+] = 10^{-\text{pH}}$, we calculated the thickness of the H^+ boundary layer at different flow rates using the pH map.

To make it clear to readers, we have added the related description in supplementary information on Page 5: “Considering $[H^+]_{\text{bulk}} = [H^+]_{z=100\mu\text{m}}$, and $[H^+] = 10^{-\text{pH}}$, we extracted the thickness of the H^+ boundary layer at different flow rates using the pH map (Fig. S26a). At flow rates of 0.16, 0.34, 0.56, and 0.77 m/s, the boundary layer thicknesses were 67, 65, 57, and 56 μm , respectively. It can be observed that the H^+ boundary layer thickness decreases as the flow rate increases. The values agree with the trend of COSOL simulations though consistently higher due to overestimation of measured surface pH.”.

Specific Comments R3-7: Time-dependent pH map should be provided to understand the dynamic evolution of vertical boundaries. Is it temporally evolving or rather stable?

Response: We appreciate the reviewer’s comments and suggestions. The system is quite stable. Under fluid flow, it rapidly reaches a steady state. For instance, using COMSOL simulations, we have shown the pH variation over time at $z=0$ under a flow velocity of 0.16 m/s, to assess when a steady state is achieved. At $t=0$ s, we introduced reactions at the anode and cathode. The system reaches a steady state within 2.5 s (Fig. S36a), which aligns closely with the convective mass transport timescale from the anode front to the cathode end (around 1.2 seconds) (Fig. S36a), defined as the distance between the anode front and the cathode end divided by the average velocity within the boundary layer. This indicates that the convective transport timescale plays a crucial role in the system’s dynamic response. During experiments, we ensure the reactions run long enough (5 minutes) to reach steady state before conducting pH measurements. While the time-dependent process by which the system achieves steady state is an interesting topic, it is beyond the scope of this paper.

Fig. S36 COMSOL simulated time-dependent pH under velocity of 0.16 m/s at (a) cathode end and (b) $z = 0 \mu\text{m}$.

To make it clear to readers, we have added a related description in the manuscript Paragraph 3 Page 26 “The COMSOL simulation shows that the pH reaches a steady state within 2.5s (Fig. S36). During experiments, we ensure the reactions run long enough (5 minutes) to reach this equilibrium before conducting pH measurements.” and added Figure S36 in the revised supplementary information.

Additional Revisions

We changed some figures and statements in the main manuscript due to some minor mistakes.